# Circadian Volume Changes in Hippocampal Glia Studied by Label-Free Interferometric Imaging

**DOI:** 10.3390/cells11132073

**Published:** 2022-06-30

**Authors:** Ghazal Naseri Kouzehgarani, Mikhail E. Kandel, Masayoshi Sakakura, Joshua S. Dupaty, Gabriel Popescu, Martha U. Gillette

**Affiliations:** 1Neuroscience Program, University of Illinois at Urbana-Champaign, Urbana, IL 61820, USA; naseri.ghazal@gmail.com; 2Beckman Institute for Advanced Science & Technology, University of Illinois at Urbana-Champaign, Urbana, IL 61820, USA; mkandel@groq.com (M.E.K.); ms4@illinois.edu (M.S.); gpopescu@illinois.edu (G.P.); 3Department of Electrical and Computer Engineering, University of Illinois at Urbana-Champaign, Urbana, IL 61820, USA; 4Department of Biomedical Engineering, Mercer University, Macon, GA 31207, USA; jdupaty@bu.edu; 5Department of Bioengineering, University of Illinois at Urbana-Champaign, Urbana, IL 61820, USA; 6Department of Cell & Developmental Biology, University of Illinois at Urbana-Champaign, Urbana, IL 61820, USA

**Keywords:** quantitative phase imaging, diurnal cycle, astrocyte dynamics, gradient light interference microscopy (GLIM)

## Abstract

Complex brain functions, including learning and memory, arise in part from the modulatory role of astrocytes on neuronal circuits. Functionally, the dentate gyrus (DG) exhibits differences in the acquisition of long-term potentiation (LTP) between day and night. We hypothesize that the dynamic nature of astrocyte morphology plays an important role in the functional circuitry of hippocampal learning and memory, specifically in the DG. Standard microscopy techniques, such as differential interference contrast (DIC), present insufficient contrast for detecting changes in astrocyte structure and function and are unable to inform on the intrinsic structure of the sample in a quantitative manner. Recently, gradient light interference microscopy (GLIM) has been developed to upgrade a DIC microscope with quantitative capabilities such as single-cell dry mass and volume characterization. Here, we present a methodology for combining GLIM and electrophysiology to quantify the astrocyte morphological behavior over the day-night cycle. Colocalized measurements of GLIM and fluorescence allowed us to quantify the dry masses and volumes of hundreds of astrocytes. Our results indicate that, on average, there is a 25% cell volume reduction during the nocturnal cycle. Remarkably, this cell volume change takes place at constant dry mass, which suggests that the volume regulation occurs primarily through aqueous medium exchange with the environment.

## 1. Introduction

Memory is encoded in the structure and function of the hippocampus [1]. The laminar structure and organization of information flow are well established. The dentate gyrus (DG) is the initial layer that receives and parses incoming signals before transmitting them to the CA3 and CA1 layers for further processing [2]. DG consists of three layers: granular, molecular, and polymorphic (hilus). The major input pathway is through the incoming axons from the entorhinal cortex that form synapses on the dendrites of the granule neurons in the molecular layer of the dentate gyrus (ML/DG). Discrete populations of astrocytes also reside in this layer [3]. This clear topographic segregation positions the astrocytes of the ML/DG to strongly influence signals coming into the DG.

It has been shown extensively that the time-of-day significantly influences hippocampal DG neuronal excitability and function [4]. Long-term potentiation induction and magnitude in the rat DG are reported to be higher during the dark phase, the active phase of the nocturnal animal [5,6,7]. Hippocampal neurogenesis in the rat DG subgranular zone also exhibits a day-night variation. The number of mitotic cells increases significantly during the night, which matches the night time increase of newborn neurons [8]. Daily oscillations have been observed in susceptibility to DG-dependent seizures. In both rat models and humans with mesial temporal lobe epilepsy (mTLE), DG neuronal excitability and seizure occurrence are time-of-day dependent, with peaks during mid-to-late daytime [9,10].

Glial cells have been implicated as major contributors to epilepsy incidence. Significant changes in glial morphology and function have been reported in different types of epilepsy, which may underlie enhanced neuronal excitability [11,12,13]. However, whether astrocyte morphologic changes occur under physiological conditions and display a diurnal pattern is still unknown. Astrocytes are a major type of glial cells. At the single-cell level, astrocytes have well-known active roles [1]. They modulate neuronal excitability and synaptic plasticity through (1) formation of tripartite synapses with pre- and post-synaptic neurons [14,15], (2) uptake or release of glutamate, ions, and neuroactive substances [16,17], (3) regulation of neuronal energy metabolism through the exchange of nutrients and metabolites with the cerebrovascular system [18,19], and (4) control of extracellular space volume [20].

Differential interference contrast (DIC) has long been used along with whole-cell patch-clamp recording [21]. However, since DIC is not a quantitative method, its applications are limited to simple observations related to cellular shape. Quantitative phase imaging (QPI) [22,23,24] is a label-free optical imaging modality that has found numerous biomedical applications as a non-destructive, quantitative alternative to traditional microscopy [25,26,27,28,29,30,31]. Recently, gradient light interference microscopy (GLIM) has been developed as a QPI technique, specifically dedicated to imaging through thick, strongly scattering specimens [32,33]. GLIM combines traditional DIC with phase-shifting interferometry and can be implemented as an upgrading optical module to an existing microscope. By controlling the phase offset between the two sheared, interfering fields in DIC, GLIM suppresses the unwanted multiple scattering background. GLIM yields the phase associated with the object’s scattering potential and has the capability to simultaneously obtain information on cell dry mass and volume in densely packed structures, such as embryos and spheroids [32,33].

Here, we quantify the diurnal vs. nocturnal temporal dynamics of astrocyte morphology in the ML/DG, utilizing the emerging technology of real-time GLIM imaging (Figure 1). We hypothesize that astrocytes of the ML/DG are morphologically dynamic over the day-night cycle. Specifically, we measure the changes in astrocyte volume and dry mass.

The advances in our study can be summarized as follows. (1) We developed an advanced investigation system that combines electrophysiology, typical DIC and fluorescence imaging channels with GLIM imaging capabilities. (2) We demonstrated the improved contrast and resolution that GLIM provides over regular DIC in acute brain slices. (3) Using software developed in-house, we achieved fully motorized GLIM data acquisition, which allows us to image entire cross-sections of acute rat brain slices, as well as z-stacks. (4) We showed that GLIM can distinguish between normal and damaged astrocytes, which are otherwise indistinguishable by patch clamp. (5) Using GLIM, we obtained astrocyte volume and dry mass with single-cell resolution in the highly dense environment of living acute brain slices. (6) Measuring astrocytes from 9 animals per each day and night group (129 cells in total), we found that there is a significant reduction in cell volume during the night. (7) Our results show that the cell volume regulation takes place at constant dry mass, which in turn indicates that the functional content of the cell remains unchanged, and the volume modulation is achieved via the exchange of aqueous fluid. (8) These results were obtained by using a non-destructive imaging technique, without the need for whole-cell patch-clamp recording and immunohistochemistry.

## 2. Results

### 2.1. Optical System

In order to study the dry mass and volume circadian changes specifically of astrocytes within the acute brain slices, we developed an optical system that augments an existing electrophysiology microscope, equipped with DIC capabilities. Figure 1a shows the actual system used in this study. The detailed optical diagram of our interferometric imaging system is depicted in Figure 1b. To reduce the scattering background from the 3–400 μm thick brain slices, we employed an LED source with emission at 780 nm. The condenser lens system and the objective itself are aligned such that the sample is illuminated by a uniform field. Before reaching the sample, the IR light passes through a polarizer and Wollaston prism, which splits the illumination into two orthogonally polarized beams. These two beams are slightly shifted transversally at the image plane, with a shift that is approximately half of the point spread function diameter at that plane. The sample scatters light that is directed back through the objective and, via the tube lens, creates a magnified replica of the sample field at the camera. Upon passing through the Wollaston prism, the two beams are now transversely overlaid but maintain their mutually orthogonal polarizations. In order to decouple the phase and amplitude information and boost the intrinsic contrast of the image, we developed a phase-shifting system that operates on GLIM’s principles [33]. Taking advantage of a transmissive, phase-only, liquid crystal variable retarder (LCVR), we created an add-on GLIM module in a very compact form factor. The LCVR controls the phase of one of the orthogonally polarized beams, specifically, the beam that has the polarization aligned with the active axis of the liquid crystal. To allow for interference at the camera, the two beams are finally passed through a polarizer oriented at 45° with respect to both polarizations to render them parallel in polarization. A personal computer synchronizes the camera acquisition with the LCVR modulation, which is precisely calibrated to add four phase shifts in increments of π/2, i.e., ε*_n_* = *n*π/2, *n* = 0, 1, 2, 3.

The resulting field at the detector is a coherent superposition of two fields, namely,
(1)$$U_{n}\left(r\right)=U\left(r\right)+U\left(r+\delta r\right)e^{i\varepsilon_{n}}$$
where $\delta r=\delta x\hat{x}$ is the spatial offset between the two fields, and *U* is the image field. The irradiance for each phase shift, $I_{n}\left(r\right)={\left|U_{n}\left(r\right)\right|}^{2}$ can be written as
(2)$$I_{n}\left(r\right)=I\left(r\right)+I\left(r+\delta r\right)+2\left|\gamma \left(r,\delta r\right)\right|\cos\left[\phi \left(r+\delta r\right)-\phi \left(r\right)+\varepsilon_{n}\right]$$
where $I\left(r\right)$ and $\phi \left(r\right)$ are, respectively, the intensity and phase of the image field, and γ is the *mutual intensity*, i.e., the temporal cross-correlation function between these two fields, evaluated at zero delay, $\gamma \left(r,\delta r\right)={\langle U^{*}\left(r\right)U\left(r+\delta r\right)\rangle }_{t}$. The term $\varepsilon_{n}=n\pi /2$ is the phase offset between the two fields, controlled by the LCVR. From the four intensity images, $I_{n},\;n=1,\ldots ,4$, we are able to solve for $I\left(r\right),$$\left|\gamma \left(r,\delta r\right)\right|$, and $\Delta \phi \left(r\right)=\phi \left(r+\delta r\right)-\phi \left(r\right).$ These data render quantitatively the gradient of the phase along the direction of the shift, $\nabla_{x}\phi \left(r\right)≃\frac{\Delta \phi \left(r\right)}{\delta x}$, by subtracting the intensities corresponding to the *cosine* and *sine* terms.
(3)$$\nabla_{x}\phi (r)=\arg\left[\frac{I_{0}(r)-I_{2}(r)}{I_{3}(r)-I_{1}(r)}\right]$$

The measured phase gradient, $\nabla_{x}\phi ,$ can be integrated along the gradient direction to get the phase map, $\phi \left(r\right)$, as described in [33].

The workflow of our investigation is presented in Figure 1c. Brains are harvested from rats separated in two “day” and “night” groups, and slices are sectioned according to standard procedures (see Materials and Methods for details). GLIM tomograms are acquired from the dentate gyrus (DG) of the hippocampus, while the brain slices are maintained viable during the experiments. To identify astrocytes specifically, we use either patch-clamp identification or colocalized fluorescence imaging. Using these labels, we segment the cells in the GLIM data and compute their volume and dry mass values.

The boost in contrast that GLIM brings over regular DIC is shown in Figure 2. It can be seen in DIC that the contrast is reduced due to multiple scattering, such that the apex of the DG is hardly visible (Figure 2a). GLIM, on the other hand, significantly reduces artifacts due to multiple scattering. Since the incoherent background is unaffected by LCVR modulation, the background introduces a common offset to the DIC intensity images and is eliminated during phase-retrieval (Equation (3), pair of frame subtractions). Interestingly, strong scattering affects not only the contrast but also the resolution. Of course, the NA of the imaging system is the same for both DIC and GLIM, which, under diffraction-limited conditions, means that the resolution should be the same. However, the scattering background noise affects the spatial high-frequency components of the DIC much more strongly than GLIM, as evidenced by the power spectra in Figure 2. As a result, the fine details in the DIC images are lost in the noise.

We developed a dedicated C++ software that controls the GLIM module, camera acquisition, and stage scanning in all three dimensions, which allows for automatic scanning of large fields of view, as well as z-stacks. Figure 3 and Appendix A illustrate the capability of our system to image entire brain cross-sections, with high contrast and submicron resolution. The individual GLIM frames that represent the tiles in the mosaic of Figure 3 were acquired at ~0.5 s each. This time includes the LVCR 4-step modulation, the 4-frame camera acquisition, and the stage translation. Because of the IR radiation and low exposure used, GLIM is nondestructive and, thus, suitable for long-term imaging of live brain slices, as illustrated in Appendix A.

### 2.2. Cell Dry Mass

The cell dry mass is obtained by summing the phase values over the segmented volume, as:(4)$$m=\frac{1}{2n_{0}\alpha }\int_{r\in V}\chi \left(r\right)d^{3}r$$

In Equation (4), χ is the scattering potential [34], $\chi \left(r\right)=n_{}^{2}\left(r\right)-n_{0}^{2}$, with n being the cell refractive index and n_0_ being the refractive index of the surrounding medium [35,36]. The coefficient α=0.2 mL/g is the typical value of the refractive index increment [37]. Interestingly, we obtained negative total dry mass values from seven cells. These cells died within two minutes from the patch, as identified by a sudden change in the Vm to highly depolarized values (Figure 4). Thus, we label these cells as “damaged” (Figure 4b). As the cellular mass presents negative values, i.e., less dense than the surrounding tissue, we conclude that these cells were essentially empty despite appearing to maintain a viable membrane potential. Alternatively, the reduction in density agrees with the expected swelling of cells during necrosis [38]. These results suggest that QPI can be used to further discriminate between morphologically damaged cells that would, at least temporarily, appear viable due to their electrophysiological signature. Whole-cell patch-clamp recording of nine astrocytes during the day and eight astrocytes during the night did not show any significant differences with regards to the electrophysiological parameters, including Vm and Rin, between day and night (Figure 4a). While patch clamping provides unique information about cellular membranes, it is a slow and tedious process that involves manipulating individual cells.

### 2.3. Cell Volume and Mass Changes during the Circadian Cycle

To improve throughput, for the bulk of this work, we also used fluorescence tagging to identify astrocytes, performing fluorescence rather than electrical identification of astrocytes (Figure 5). This way, we acquired between one to six brain slices per animal per timepoint, with a total of 45 brain slices (20 night, 25 day), from 18 animals (9 night, 9 day). From these tomograms we annotate a total of 129 cells with 135 z-slices per cell. As outlined in materials and methods, we coarsely segmented the tomograms to obtain a mask of the cell body, giving us access to the cellular dry mass within the cell body as well as the volumetric information.

Our measurements indicated that during the nocturnal portion of the circadian cycle, the cellular bodies were, on average, 25% smaller in volume compared to the diurnal counterpart (Figure 6a,b). Interestingly, these volume changes occur at constant dry mass, i.e., we found statistically insignificant dry mass differences between the two groups (Figure 6c,d). The volume vs. mass slopes (Figure 6e,f) are 1.02 × 10^−3^ L/g and 1.17× 10^−3^ L/g for Day and Night, respectively. These numbers lead to density values of 980 g/L and 850 g/L. We note that these values are only slightly lower than expected for mammalian cells and are compatible with our previous results in HeLa using GLIM [33]. The slight underestimation of mass values is likely due to the dense background surrounding the cells in the brain slice. In addition to optical effects, it is known that density can vary depending on cell type, where volume regulation is likely mediated mainly by water exchange with the environment [39]. Thus, while the absolute measurements of cellular mass are only comparable to values obtained for similar samples behaving in similar ways, the relative differences, analyzed here, may help gain a better understanding of astrocyte function, especially in the context of neurogenesis and degeneration [40].

## 3. Discussion

This work presented a method for analyzing cell dry mass and volume information within strongly scattering samples, such as acute brain slices. Thus, we were able to observe cellular level detail using GLIM microscopy, and we developed a 3D annotation scheme that uses phase information. Furthermore, we integrated GLIM imaging on an electrophysiology system, while maintaining the patch-clamp and fluorescence imaging capabilities intact. The image contrast in GLIM is significantly improved over the DIC due to the significant reduction of the sample-induced, inhomogeneous, multiple scattering background. Importantly, this background also affects the high-spatial frequency content of the DIC images, thus reducing the spatial resolution. Thanks to the software automation developed in our laboratory, we were able to acquire cross-sections of entire brain slices consisting of 1600 frames in less than 15 min. Compared to fluorescence imaging, GLIM requires considerably less power density at the sample plane. Thus, using IR full-field illumination and low exposure, we were able to image live brain slices over many hours, while the viability remained limited by the environmental conditions rather than the imaging itself (Appendix A shows a movie over 20 h).

While there was a significant difference in astrocyte cell body volume between the day and nighttime, we found no statistically meaningful change in cellular dry mass. As dry mass represents mainly lipids and proteins, the lack of a significant circadian difference in the dry mass suggests that the amount of lipids and proteins is maintained over the day-night cycle. One probable explanation for these observations is that the change in astrocyte volume is due to a change in the water and ion content from day to night, rather than synthesis followed by elimination of new cellular material. Astrocytes have been shown to extend and retract fine processes across numerous physiological states such as the day-night cycle [41,42], sleep [43], pregnancy [44,45], and dehydration [46]. It is thought that astrocytes retract their processes allowing for an increase in the volume of extracellular space during sleep, which in turn helps with waste clearance [47].

Changes in astrocyte volume often occur under conditions of high extracellular potassium concentrations, such as during increased neuronal activity. Astrocytic swelling is mediated by fluxes of ions, and subsequent movement of osmotic-dependent water occurs by a diverse set of volume sensitive ion channels and transporters reviewed in [48], although likely independent of the astrocytic-specific aquaporin 4 (AQP4) channel [49].

Although astrocytes undergo dramatic volume changes during pathological conditions [50,51], they have also exhibited volume adjustments under physiological conditions. Real-time measurements of astrocytic volume changes performed under 2-photon imaging revealed dynamic volume changes in response to physiological increase in extracellular potassium as well as GABA_A_ receptor activation [52]. Additionally, using capacitance measurements, astrocytic volume changes were demonstrated during slow sleep oscillations and seizures, tightly related to neuronal activation [53].

Our finding demonstrating increased cell body volume during the day, the sleep phase for these nocturnal animals, is in agreement with studies showing a decrease in astrocytic fine processes coverage of synapses during sleep [43], and with studies that demonstrate a reduction in interstitial space measurements during wakefulness [47]. Overall, this may suggest a role of astrocyte morphological dynamics in modulating synaptic transmission and diurnal physiology or sleep state. We note that, while this work focused on the cell bodies of astrocytes, we expect the approach presented here to be generally applicable to imaging other cell types found in the brain, such as various types of neurons and glia.

## 4. Materials and Methods

### 4.1. Animals

Long-Evans/BluGill rats of 3–6 weeks of age were used for this study. Breeding colonies were generated and maintained at the University of Illinois at Urbana-Champaign (UIUC). Animals were housed under standard conditions on a 12:12 h L/D schedule and given food and water ad libitum. All experimental protocols complied with the National Institutes of Health Public Health Service Policy on Humane Care and Use of Laboratory Animals and were approved by the UIUC Institutional Animal Care and Use Committee. We note that, unlike humans, rats are nocturnal and are most active during the night.

### 4.2. Brain Slice Preparation and Patch-Clamp Recording

Animals were sacrificed either at the onset of light for early day recordings or right before the offset of light for early night recordings. Coronal rat hippocampal slices of 300 µm thickness were prepared as described previously [54]. Briefly, the brain was quickly removed and placed into an ice-cold slicing solution saturated with 95% O_2_/5% CO_2_ and was cut using a vibrating tissue slicer. Brain slices were then placed into a holding chamber containing artificial cerebrospinal fluid (ACSF) saturated with 95% O_2_/5% CO_2_ at room temperature. The slices were incubated for an hour before recording.

Astrocytes were identified by (1) location and morphology using GLIM and (2) electrophysiological properties, including lack of electrical excitability, highly negative resting membrane potential (Vm), and low resistance as assessed by the current-voltage (I-V) curves. Intracellular recordings of astrocytes, using the whole-cell patch-clamp technique, were obtained by electrodes with pipette-tip resistances of 7–10 MΩ. These microelectrodes were filled with an intracellular solution, as described previously [54]. Under current-clamp mode, the Vm of the astrocytes was recorded at either early day or early night. The input resistance (Rin) of the cells was measured from the linear slope of the I-V curve, obtained by applying a current-step protocol (duration 600 ms) from −100 pA to +200 pA with 20 pA increments. A MultiClamp 700B amplifier was used for current-clamp recordings (Molecular Devices, San Jose, CA, USA). Data were stored on a computer for further analyses using the pClamp software (Molecular Devices, San Jose, CA, USA).

### 4.3. Colocalized Fluorescence and GLIM Imaging

In this work, we used the compact GLIM module shown in [32] with a transmission IR-DIC path. Data were acquired with an Axio Examiner (ZEISS, Jena, Germany) and Zyla (Andor Technology, Oxford Instruments, Abingdon, UK) camera at 63×/1.0 (VIS-IR, ZEISS, Jena, Germany). Whole-brain slide scan was performed with an Axio Observer Z1 (ZEISS, Jena, Germany) with a 40×/0.75 objective and motorized stage (MAC5000, Ludi Electronic Products (LEP), Hawthorne, NY, USA). A fully open condenser (NA = 0.55) was used for all images, affording a lateral resolution of 0.75 μm and 1.5 μm axially [32]. Fluorescence microscopy was used for the identification of astrocytes. A piezo focus scanning objective (PD72Z1SAQ, Physik Instrumente, Auburn, MA, USA) was used to sequentially acquire colocalized fluorescence and GLIM tomograms. For each field of view, we obtain two tomograms of 175 × 175 × 27 µm which typically contain several astrocytes.

Solfurhodamine 101 (SR101) is a water-soluble compound, widely used for preferential labeling of astrocytes for in vitro microscopy experiments [55] which leads to labeling of astrocyte cell bodies and proximal processes [56,57]. The uptake of SR101 by astrocytes is due to the presence of an organic anion-transporting polypeptide called OATP1C1 [58]. Although OATP1C1 is expressed preferentially in astrocytes, it is also found to be expressed in oligodendrocytes [59,60] as well as in neurons [61]. The reason we can claim with high confidence that we are only visualizing astrocytes in the ML/DG using this technique is that: (1) oligodendrocytes do not reside in this hippocampal layer, (2) neurons can only uptake the dye at very high concentrations, 165 µM, while we are using a concentration of only 1 µM, and (3) neurons have been shown to de-stain very quickly, while the protocol requires an hour of wait time after incubation before the start of recording [60].

A modified low-Ca^2+^ high-Mg^2+^ ACSF was prepared according to protocols for animals older than 15 days [56,57]. This modified ACSF contained NaCl 125 mM, KCl 2.5 mM, NaH2PO_4_ 1.25 mM, NaHCO_3_ 26 mM, CaCl_2_ 0.5 mM, MgCl_2_ 6 mM, and glucose 20 mM, 300 mOsm/L, saturated with 95% O_2_/5% CO_2_. Live brain slices were incubated in modified ACSF containing 1 µm SR101 (Thermo Fisher Scientific, Waltham, MA, USA) at 34 °C for 20 min, followed by incubation in normal ACSF for 10 min at 34 °C to allow washout of excess dye from the extracellular space. Brain slices were then kept in normal ACSF at room temperature until they were used for imaging.

Slices prepared for patch-clamp recording were prepared in a similar fashion less the stain, and were analyzed together with labeled cells. In this case, dry mass was measured before patching.

### 4.4. Analysis of GLIM Data

To obtain the phase shift associated with the object, we performed phase retrieval as outlined in [32] followed by a Hilbert transform approach for numerical integration [62]. In order to measure the dry mass of individual cells, we performed a coarse manual annotation in 3-dimensional space, which was then automatically refined into a high-fidelity segmentation mask. Specifically, the outline of the cell was traced in a small number of z-slices using the ImageJ software’s ROI drawing capabilities (ImageJ, NIH, Bethesda, MD, USA). In addition to providing quantitative information, phase-shifting holistically improves image quality by suppressing incoherent multiple scattering contributions, making it easier to annotate and analyze the images. Using a purpose-built tool written in MATLAB (Natick, MA, USA), we converted the point cloud from the tracings into a volume using a Delaunay Triangulation. Next, the triangulation was converted into a 3D binary mask by testing whether voxels lay within the mesh (code available at [63]). To further refine this mask, we distinguished between background and cellular material by noting that the unwanted background has a low phase shift near zero. We stress that this last step, the refinement of the mask using quantitative phase information, is only possible with interferometric hardware.

To obtain the dry mass we accumulated the non-background (positive) phase inside this mask, with the number of voxels corresponding to the volume. We note that our technique preferentially annotates cellular bodies rather than extensions—the latter are often obscured by neighboring cells.

### 4.5. Statistical Analysis

Statistical analysis was performed using SAS statistical software. All data are presented as mean ± SEM. Mann-Whitney *t*-test was used due to the non-normality of the data. Statistical analysis was performed using SAS statistical software (SAS Studio, SAS Institute Inc., Cary, NC, USA). All data are presented as mean ± SEM. Due to the nature of measurements representing multiple slices/animals/time points, a one-way mixed model ANOVA was used with Tukey’s post-hoc comparison. The fixed, between-subject factor was the time-of-day with 2 levels: early day and early night. The random within-subject factor was the individual animal to account for the correlation between the multiple slices within each individual as well as the variation between different animals. *p*-values < 0.05 were considered statistically significant.

## Figures and Tables

**Figure 1 cells-11-02073-f001:**
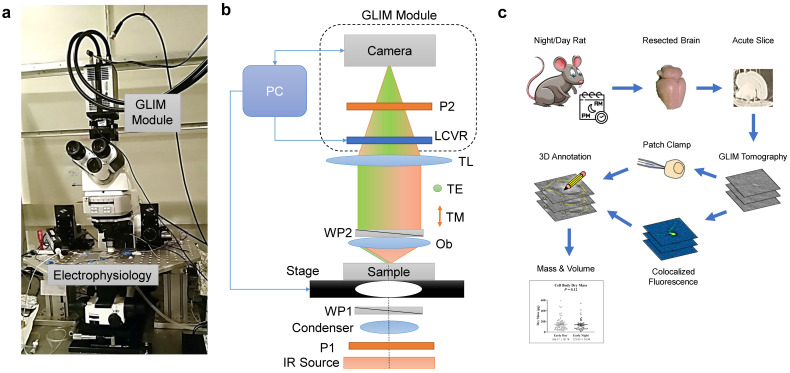
**Gradient light interference microscopy and electrophysiology system to characterize acute brain slices.** (**a**) Compact GLIM module attached to a motorized microscope equipped with infrared DIC illumination and electrophysiology patch-clamp system. (**b**) Interferometric imaging system. An IR source is uniformly illuminating the sample via a polarized condenser. The Wollaston prism WP1 splits the unpolarized light into TE and TM polarizations, which are slightly shifted at the sample plane. The light scattered from the sample passes through the WP2 such that the two beams are now recombined. The tube lens (TL) forms a magnified image at the camera plane. Before reaching the detector, the liquid crystal variable retarder (LCVR) modulates the phase of one polarization with respect to the other. An output polarizer (P2) projects the orthogonal polarizations to the same direction such that they interfere at the camera plane. A personal computer (PC) synchronizes precisely the LVCR modulation with camera acquisition and stage position. (**c**) GLIM was used to study circadian changes in astrocyte dry mass and morphology. Acute brain slices were harvested from rats during the day or night portion of the cycle. GLIM tomography was performed followed by patch clamp and fluorescence confirmation of astrocytes. Data were annotated using a 3D software tool developed in-house.

**Figure 2 cells-11-02073-f002:**
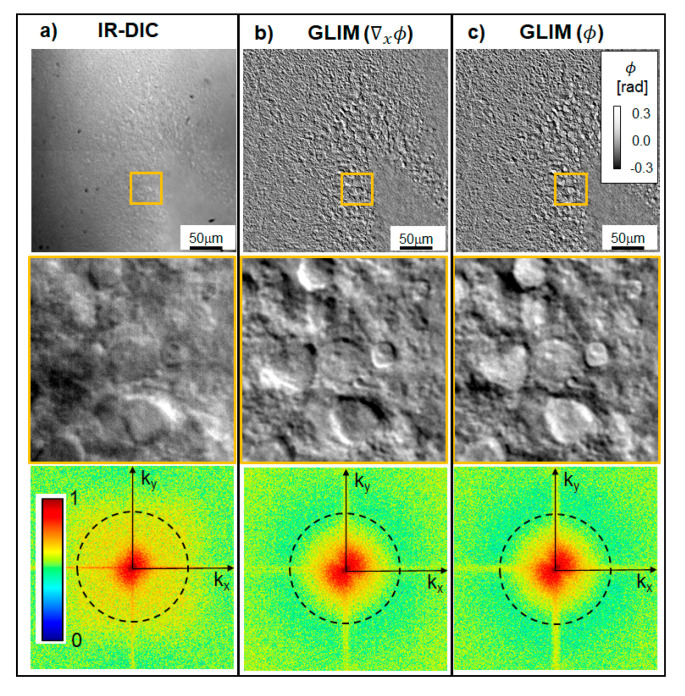
**GLIM improves image contrast and resolution of a 300 µm thick acute brain slice by decoupling phase from unwanted amplitude information**. (**a**) Typical DIC image using IR illumination. (**b**) Quantitative phase gradient ($\nabla_{x}\phi$) map. (**c**) Quantitative phase image, ϕ, obtained by Hilbert transform integration. Orange boxes depict zoomed-in regions of interest in the granular layer of the dentate gyrus (DG). The bottom row shows the spatial power spectra associated with the respective zoomed-in images. The DIC image is characterized by a narrower power spectrum due to the scattering background noise corrupting the high-frequency range. As a result, the DG apex, cellular and subcellular structures are more clearly visible in GLIM. The dash rings represent the diffraction limit (40×/0.75NA objective). The power spectra are plotted in logarithmic scale, normalized to unit maximum.

**Figure 3 cells-11-02073-f003:**
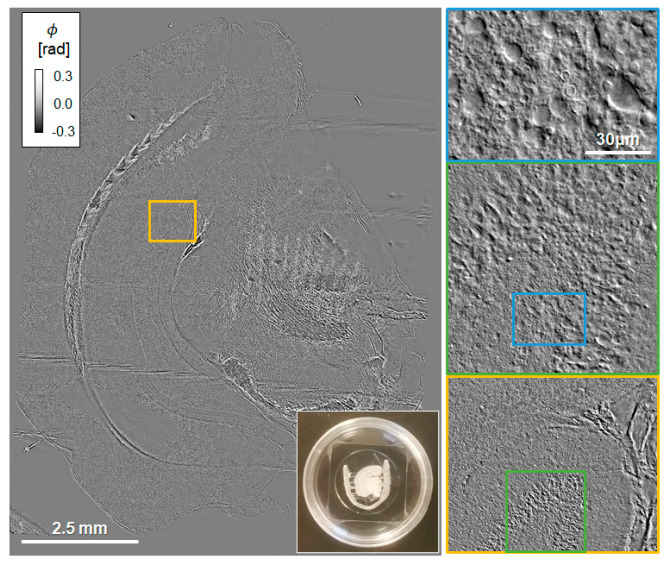
**GLIM image of an entire acute brain slice cross-section.** The pairs of colored rectangles illustrate zoomed-in regions of interest. The image represents a 40 × 40 tile mosaic, acquired in 15 min (40×/0.75NA objective).

**Figure 4 cells-11-02073-f004:**
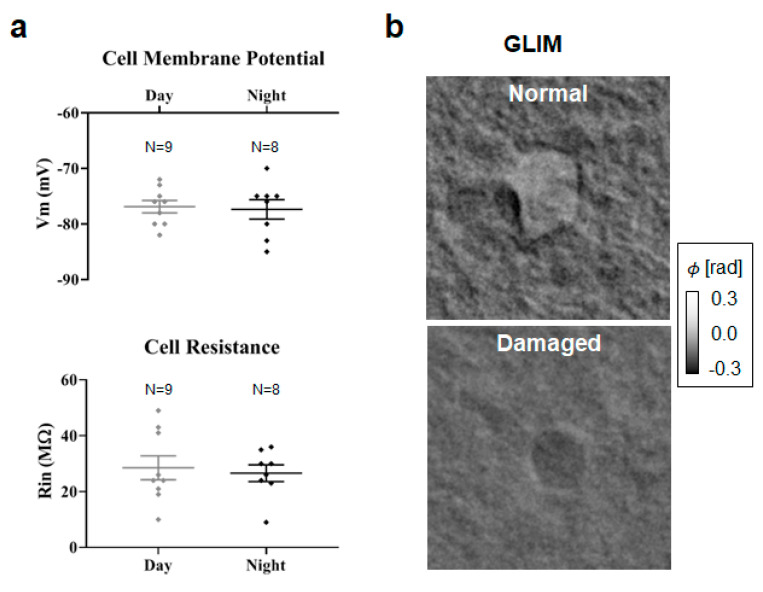
**Astrocytes during the circadian cycle are indistinguishable by electrophysiology.** (**a**) Comparison of day vs. night cells shows no statistically significant difference in membrane potential and resistance values. (**b**) GLIM reports different phase maps for normal and damaged cells (63×/1.0NA objective).

**Figure 5 cells-11-02073-f005:**
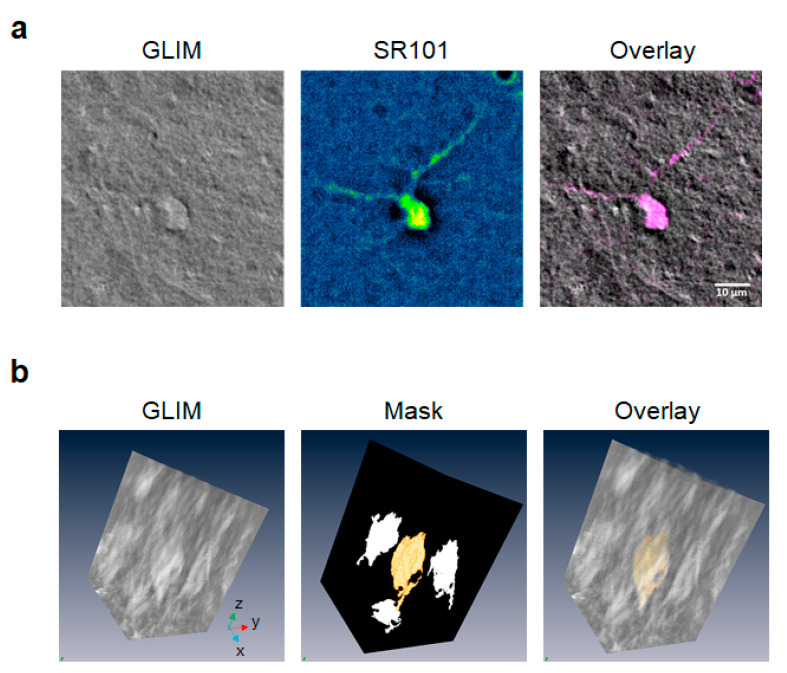
**Sulforhodamine 101 enables identification of cells imaged by GLIM as astrocytes.** (**a**) Live brain slices were incubated in ACSF containing 1 µm Sulforhodamine 101 (SR101). SR101-positive astrocytes were identified using fluorescent microscopy and z-stack images of the region of interest were acquired (63×/1.0NA objective). The colocalization of the fluorescent and GLIM images of the same ROI was used to obtain 3D measurements, including cell volume and dry mass. (**b**) Dry mass maps were obtained by a coarse annotation of the GLIM volume, followed by computational, threshold-based refining of the course map. The resulting segmentation masks obtained from fluorescence were used on the GLIM channel to extract the dry mass and cellular volume.

**Figure 6 cells-11-02073-f006:**
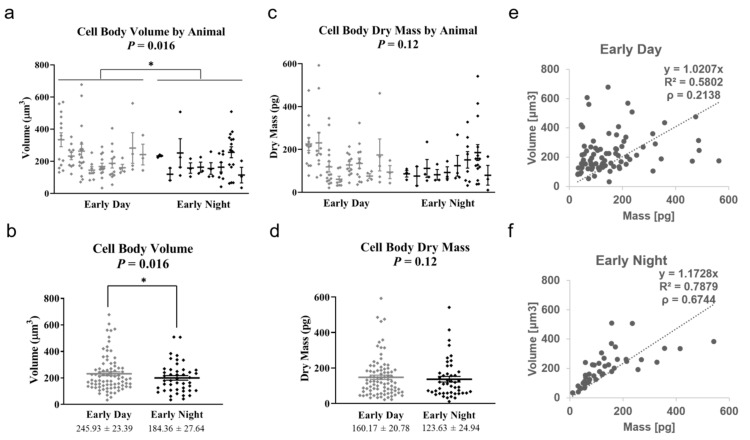
**Astrocyte cell bodies display significantly larger volume during the day vs. night time.** (**a**) Distribution of cell body volumes, with cells for individual animals shown as points on separate vertical lines. * *p* < 0.05 (**b**) Cell volume distribution obtained by combining measurements from all animals. The volume distributions for day vs. night show statistical significance. * *p* < 0.05 (**c**) Distribution of cell dry mass values, with cells for individual animals shown as points on separate vertical lines. (**d**) Cell dry mass distribution obtained by combining measurements from all animals. The dry mass distributions for day vs. night show no statistical significance. n = 9 animals per time point (total of 129 cells). Error bars represent standard error of the mean (SEM). (**e**,**f**) show that the correlation between dry mass and volume is more pronounced during the early night (**f**).

## Data Availability

The data that support the findings of this study are available upon reasonable request.

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
