# Peer review of "Circadian Volume Changes in Hippocampal Glia Studied by Label-Free Interferometric Imaging"

_cells, 2022, doi:10.3390/cells11132073_

Round 1
Reviewer 1 Report
Kouzehgarani et al., present an interesting report. Using label-free interferometric imaging, volume changes in hippocampal glia were systematically studied. In particular, they utilized gradient light interference microscopy (GLIM) for label-free quantitative phase imaging of thick brain tissues and electrophysiology to 25 quantify the astrocyte morphological behavior over the day-night cycle. The experiments must be very challenging, but the authors nicely demonstrated the observation of circadian volume changes. The manuscript is well motivated and clearly written. I recommend the publication of this manuscript. The followings are optional suggestions to improve the quality of the manuscript.
(major)
Fig. 6: Can the authors also report the dry mass concentrations? I think it can be calculated from the mean refractive index values and the known refractive index increment value. In addition, I would like to suggest including the scatter plots between dry mass vs. volume – this might provide information on the potential dependence of the circadian cycle on biophysical parameters.
(minor)
Figs. 2,3, and 4: please add colormaps for both GLIM images and Fourier spectra.
Section 2. Results: I would like to suggest including subsection titles.
Fig. 6: The background colors are distracting. I would remove the backgrounds.
Author Response
Kouzehgarani et al., present an interesting report. Using label-free interferometric imaging, volume changes in hippocampal glia were systematically studied. In particular, they utilized gradient light interference microscopy (GLIM) for label-free quantitative phase imaging of thick brain tissues and electrophysiology to quantify the astrocyte morphological behavior over the day-night cycle. The experiments must be very challenging, but the authors nicely demonstrated the observation of circadian volume changes. The manuscript is well motivated and clearly written. I recommend the publication of this manuscript. The followings are optional suggestions to improve the quality of the manuscript.
We thank the reviewer for correctly summarizing our manuscript and for the encouraging comments. We address each comment bellow and mention the associated revisions.
(major)
Fig. 6: Can the authors also report the dry mass concentrations? I think it can be calculated from the mean refractive index values and the known refractive index increment value. In addition, I would like to suggest including the scatter plots between dry mass vs. volume – this might provide information on the potential dependence of the circadian cycle on biophysical parameters.
Following the reviewer suggestion, we now add the scatter plots between dry mass vs. volume in new panels for Fig. 6 (e &f).
(minor)
Figs. 2,3, and 4: please add colormaps for both GLIM images and Fourier spectra.
Thank you, we now added color bars to all Figures.
Section 2. Results: I would like to suggest including subsection titles.
In the revised version, we now add subsections for clarity.
Fig. 6: The background colors are distracting. I would remove the backgrounds.
Following the reviewer suggestion, we now removed the background.
Reviewer 2 Report
The authors proposed a very interesting technique, good control of the nature of the cells they analyzed. GLIM microscopy looks promising. The phenomenon of changes in the volume of anthracites in the day-night cycle is also interesting, but it was possible to discuss the biological significance and possible causes in more detail. In general, certain groups of ion channels can affect the volume, this should be mentioned in the discussion, which in my opinion is too concise. An interesting phenomenon has been obtained, but almost not discussed in any way. Just shown as is. It is not discussed in which other cases changes in cell volume are described in the literature. And there are not so many of them and could be compared with the results obtained by the authors.
According to the methodology, everything is fine, but they should have paid more attention to monitoring the state of cells. Due to the effect on the volume, it was possible to try to filter out cells potentially in a borderline state between life and death. Since the patch clamp on the surviving sections always carries a risk to the condition of the cells.
Some of them are in unfavorable conditions and generally live only for a certain time. In this regard, it would be interesting to know how the authors control the state of cells and protect the material from artifacts. Why it was not considered necessary to use specific cell degeneration dyes after lifetime experiments.
Author Response
The authors proposed a very interesting technique, good control of the nature of the cells they analyzed. GLIM microscopy looks promising.
We thank the reviewer for the encouraging comments and for the valuable feedback.
- The phenomenon of changes in the volume of astrocytes in the day-night cycle is also interesting, but it was possible to discuss the biological significance and possible causes in more detail. In general, certain groups of ion channels can affect the volume, this should be mentioned in the discussion, which in my opinion is too concise. An interesting phenomenon has been obtained, but almost not discussed in any way. Just shown as is. It is not discussed in which other cases changes in cell volume are described in the literature. And there are not so many of them and could be compared with the results obtained by the authors.
We agree that the changes in volume are interesting and have added a discussion of the biological significance and causes in more detail to the discussion. Thank you for this suggestion.
- According to the methodology, everything is fine, but they should have paid more attention to monitoring the state of cells. Due to the effect on the volume, it was possible to try to filter out cells potentially in a borderline state between life and death. Since the patch clamp on the surviving sections always carries a risk to the condition of the cells.
We did monitor the health of the cells via electrophysiological measurements. The cells response and measurements of their membrane voltage and resistance were within established ranges.
- Some of them are in unfavorable conditions and generally live only for a certain time. In this regard, it would be interesting to know how the authors control the state of cells and protect the material from artifacts. Why it was not considered necessary to use specific cell degeneration dyes after lifetime experiments.
We did not utilize specific cell degeneration dyes due to the fact that we were measuring the health of the cells by electrophysiological measurements. We and many others have published based on the electrophysiolgical measurements of the patched cell as an indicator of their health.
Reviewer 3 Report
This manuscript comes from, or with participation, of one of the major QPI research labs, and for this reason alone it deserves every attention. Moreover, the authors tackle the difficult and important problem of imaging individual cells in the brain.
That said, I have a few comments to make:
- If the average dry mass of an astrocyte is 160*10-12 g and its average volume is 240 um3 = 240*10-15 l (as in Fig. 6b-d), that gives the dry density of 670 g/L. This value is three times larger than would be expected for a nucleated mammalian cell. I suggest that the authors at least discuss this result at length. It might also be useful to plot the dry mass vs volume for individual cells to better appreciate the slope.
- I assume GLIM achieves some depth discrimination. What would be the estimate for z resolution using an NA 0.75 objective? Has the correction for refractive index differences been applied to z stacks?
- The other reason to inquire about depth discrimination is the pockmarked appearance of the background (Fig. 3). What is the nature of the background against which the astrocytes are measured? Is it pure cerebrospinal fluid or could the background be affected by nearby cells, including cells lying out of focus?
- The observation of cells with negative mass is curious but, typically, dead cells are not empty, and it is unclear how they can maintain a viable (how much?) membrane potential. I think the interpretation of cells with negative mass should be done more cautiously.
- Has anyone applied QPI to brain imaging before?
Author Response
This manuscript comes from, or with participation, of one of the major QPI research labs, and for this reason alone it deserves every attention. Moreover, the authors tackle the difficult and important problem of imaging individual cells in the brain.
We thank the reviewer for acknowledging the challenges associated with these experiments. We address each comment below individually.
That said, I have a few comments to make:
- If the average dry mass of an astrocyte is 160*10-12 g and its average volume is 240 um3 = 240*10-15 l (as in Fig. 6b-d), that gives the dry density of 670 g/L. This value is three times larger than would be expected for a nucleated mammalian cell. I suggest that the authors at least discuss this result at length. It might also be useful to plot the dry mass vs volume for individual cells to better appreciate the slope.
We thank the reviewer for pointing out this result that requires more discussion. We now plot the volume vs. mass in Fig. 6. The volume vs. mass slope is 1.02e-3 l/g and 1.17e-3 l/g for Day and Night, respectively. These numbers lead to density values of 980 g/l and 850 g/l. We note that these values are only slightly lower than expected for mammalian cells, such as HeLa or fibroblasts: » What is the density of cells? (bionumbers.org). Our measurements are comparable with our previous results in heLa using Glim (Ref. 33). The slight underestimation of mass values is likely due to the dense background surrounding the cells in the brain slice. We comment on these results on page 10.
I assume GLIM achieves some depth discrimination. What would be the estimate for z resolution using an NA 0.75 objective? Has the correction for refractive index differences been applied to z stacks?
We thank the reviewer for allowing us to clarify the depth resolution (page 14, “affording a lateral resolution of…”).
- The other reason to inquire about depth discrimination is the pockmarked appearance of the background (Fig. 3). What is the nature of the background against which the astrocytes are measured? Is it pure cerebrospinal fluid or could the background be affected by nearby cells, including cells lying out of focus?
Although the sample is, indeed, surrounded by artificial cerebrospinal fluid (aCSF), the images are acquired within the sample where the cells are measured in their natural environment of brain tissue. As a result, the background is strongly inhomogeneous due to the surrounding cells, in addition to fluid. We now discuss further this point that not made sufficiently clear (page 10, “significant reduction of the sample-induced, inhomogeneous, multiple scattering background”).
- The observation of cells with negative mass is curious but, typically, dead cells are not empty, and it is unclear how they can maintain a viable (how much?) membrane potential. I think the interpretation of cells with negative mass should be done more cautiously.
We agree that this point benefits from a discussion in more detail. Damaged cells exhibit dry mass content below the surrounding background and, as such, report negative values. This is likely due to loss of dry content that was transported away for the cell as well as the expected swelling of the cell during necrosis. We discuss this interpretation further on page 9 (“These results suggest that QPI…”).
- Has anyone applied QPI to brain imaging before?
To our knowledge, this is the first time QPI was applied to acute brain slices. We now mention this in the Summary section.
http://book.bionumbers.org/what-is-the-density-of-cells/
For reports about 1000 g/ml for cells. Which agrees with our discussion about GLIM's underestimation due to inhomogeneous background - and is the opposite conclusion of the comment from the reviewer.
Round 2
Reviewer 3 Report
I appreciate the effort of the authors to make the requested improvements. Unfortunately, two of my main concerns about the quantitative validity of data have not been dispelled.
This manuscript comes from, or with participation, of one of the major QPI research labs, and for this reason alone it deserves every attention. Moreover, the authors tackle the difficult and important problem of imaging individual cells in the brain.
We thank the reviewer for acknowledging the challenges associated with these experiments. We address each comment below individually.
That said, I have a few comments to make:
- If the average dry mass of an astrocyte is 160*10-12 g and its average volume is 240 um3 = 240*10-15 l (as in Fig. 6b-d), that gives the dry density of 670 g/L. This value is three times larger than would be expected for a nucleated mammalian cell. I suggest that the authors at least discuss this result at length. It might also be useful to plot the dry mass vs volume for individual cells to better appreciate the slope.
We thank the reviewer for pointing out this result that requires more discussion. We now plot the volume vs. mass in Fig. 6. The volume vs. mass slope is 1.02e-3 l/g and 1.17e-3 l/g for Day and Night, respectively. These numbers lead to density values of 980 g/l and 850 g/l. We note that these values are only slightly lower than expected for mammalian cells, such as HeLa or fibroblasts: » What is the density of cells? (bionumbers.org). Our measurements are comparable with our previous results in heLa using Glim (Ref. 33). The slight underestimation of mass values is likely due to the dense background surrounding the cells in the brain slice. We comment on these results on page 10.
To my understanding, the numbers cited at http://book.bionumbers.org/what-is-the-density-of-cells/ are wet density and Fig. 6 shows dry density. Dry density in most cell types is close to 200 g/ml, i.e. five times less
I assume GLIM achieves some depth discrimination. What would be the estimate for z resolution using an NA 0.75 objective? Has the correction for refractive index differences been applied to z stacks?
We thank the reviewer for allowing us to clarify the depth resolution (page 14, “affording a lateral resolution of…”).
- The other reason to inquire about depth discrimination is the pockmarked appearance of the background (Fig. 3). What is the nature of the background against which the astrocytes are measured? Is it pure cerebrospinal fluid or could the background be affected by nearby cells, including cells lying out of focus?
Although the sample is, indeed, surrounded by artificial cerebrospinal fluid (aCSF), the images are acquired within the sample where the cells are measured in their natural environment of brain tissue. As a result, the background is strongly inhomogeneous due to the surrounding cells, in addition to fluid. We now discuss further this point that not made sufficiently clear (page 10, “significant reduction of the sample-induced, inhomogeneous, multiple scattering background”).
- The observation of cells with negative mass is curious but, typically, dead cells are not empty, and it is unclear how they can maintain a viable (how much?) membrane potential. I think the interpretation of cells with negative mass should be done more cautiously.
We agree that this point benefits from a discussion in more detail. Damaged cells exhibit dry mass content below the surrounding background and, as such, report negative values. This is likely due to loss of dry content that was transported away for the cell as well as the expected swelling of the cell during necrosis. We discuss this interpretation further on page 9 (“These results suggest that QPI…”).
This brings back the question of the background (see previous question). If the background is so high that semi-viable cells appear empty or having negative mass, something must be wrong there.
Author Response
Reviewer 3 still has an issue with the "dry weight" determination and the values cited in comparison to other measurements in the field.
We appreciate the chance to discuss further the dry mass measurements. At the same time, we would like to emphasize that absolute values are secondary to relative ones. In other words, even if we are, let’s say, off by 10%, it is much more important that the measurements for the two groups are consistent, i.e., precision is crucial over accuracy for biological measurements. For example, it is known that in the literature researchers use a refractive index increment, which converts phase images into dry mass density maps, within the range 0.18-0.21 ml/g1. Irrespective of the value chosen, dry mass measurements are informative when reporting on relative changes in cell growth, for example. We discuss some of the points further in the revised manuscripts. (Page 10 , “Thus, while the absolute measurements of cellular mass are only comparable to values obtained for similar samples behaving in similar ways, the relative differences, …”)
The authors must address this point further.
e.g.
-is this a methodological under/overestimate due to specific algorithms that the authors use in this study?
The volume vs. mass slopes (Figs. 6e-f) are 1.02e-3 l/g and 1.17e-3 l/g for Day and Night, respectively. These numbers lead to density values of 980 g/l and 850 g/l. We note that these values are only slightly lower than expected for mammalian cells and are compatible with our previous results in HeLa using GLIM33. The potential slight underestimation of mass values can be due to the dense background surrounding the cells in the brain slice. On the other hand, note that we used an average refractive index of 0.2 ml/g, within the range of accepted values of 0.18-0.21 ml/g. If we use 0.18 ml/g, the dry mas values go up by 11%. This leads to an important point in dry mass measurements: the absolute values are secondary to the relative ones. Since the precision of our GLIM measurements is extremely high, due to its common-path stability, the conclusions of our study regarding day vs. night cell behavior is not sensitive to the absolute values of dry mass density.
(Page 10 , “Thus, while the absolute measurements of cellular mass are only comparable to values obtained for similar samples behaving in similar ways, the relative differences, …”)
-could apparent difference be due to unusual nature of astrocytes? if so, are there comparable measurements of astrocytes in vitro, or perhaps of neurons slices by these authors?
We do not believe that astrocyte morphology would introduce systematic errors in GLIM. Because of the secondary role of the absolute values for this study, performing astrocyte measurements in vitro are beyond the scope of this study. It is also very likely that such a model will not mimic the properties of the cell within the brain slice, which defeats the purpose of such a control study.
Supplementary video 2: This movie seems to show extensive cell death in DG during the time lapse, and so does not support the statement on page 6 that "GLIM is non-destructive" (though it may also not show that it is). This should be removed.
The potential cell death exhibited in the brain slice is due to the environmental conditions that were not maintained. The main purpose of the movie is to show that GLIM is able to reveal cellular and subcellular structures in an unlabeled brain slice, which we believe is reported here for the first time. We also note that in our study, each cell was diagnosed for viability using electrophysiology, so there is no risk of contaminating our conclusions.
(Page 11 , “while the viability remained limited by the environmental conditions rather than imaging itself (Supplemental Video 2 shows a movie over 20 hours).”)
1 Popescu, G. et al. Optical imaging of cell mass and growth dynamics. American Journal of Physiology-Cell Physiology 295, C538-C544, doi:10.1152/ajpcell.00121.2008 (2008).